# Technical Performance, Overall Accuracy and Complications of EUS-Guided Interventional Procedures: A Dynamic Landscape

**DOI:** 10.3390/diagnostics12071641

**Published:** 2022-07-05

**Authors:** Irina Florina Cherciu Harbiyeli, Alina Constantin, Irina Mihaela Cazacu, Daniela Elena Burtea, Elena Codruța Gheorghe, Carmen Florina Popescu, Nona Bejinariu, Claudia Valentina Georgescu, Daniel Pirici, Bogdan Silviu Ungureanu, Cătălin Copăescu, Adrian Săftoiu

**Affiliations:** 1Research Center of Gastroenterology and Hepatology Craiova, University of Medicine and Pharmacy Craiova, 200349 Craiova, Romania; cherciuirina@gmail.com (I.F.C.H.); irina.cazacu89@gmail.com (I.M.C.); constantinescu.codruta@yahoo.com (E.C.G.); cfpopescu67ro@gmail.com (C.F.P.); cgeorgescu2001@yahoo.com (C.V.G.); argonautro@yahoo.com (D.P.); boboungureanu@gmail.com (B.S.U.); adriansaftoiu@gmail.com (A.S.); 2Gastroenterology Department, Ponderas Academic Hospital, 014142 Bucharest, Romania; drconstantinalina@gmail.com (A.C.); catalin.copaescu@ponderas-ah.ro (C.C.); 3Faculty of Medicine, Titu Maiorescu University, 040441 Bucharest, Romania; 4Santomar Oncodiagnostic, Regina Maria Histopathology Laboratory, 400350 Cluj Napoca, Romania; nonarebei@yahoo.com

**Keywords:** endoscopic ultrasound, endoscopic ultrasound-guided fine needle aspiration/biopsy

## Abstract

Endoscopic ultrasound (EUS) gained wide acceptance as the diagnostic and minimally invasive therapeutic approach for intra-luminal and extraluminal gastrointestinal, as well as various non-gastrointestinal lesions. Since its introduction, EUS has undergone substantial technological advances. This multi-centric study is a retrospective analysis of a prospectively maintained database of patients who underwent EUS for the evaluation of lesions located within the gastrointestinal tract and the proximal organs. It aimed to extensively assess in dynamic the dual-center EUS experience over the course of the past 20 years. Hence, we performed a population study and an overall assessment of the EUS procedures. The performance of EUS-FNA/FNB in diagnosing pancreatic neoplasms was evaluated. We also investigated the contribution of associating contrast-enhanced ultrasound imaging (CE-EUS) with EUS-FNA/FNB for differentiating solid pancreatic lesions or cystic pancreatic lesions. A total of 2935 patients undergoing EUS between 2002–2021 were included, out of which 1880 were diagnostic EUS and 1052 EUS-FNA/FNB (80% FNA and 20% FNB). Therapeutic procedures performed included endoscopic transmural drainage of pancreatic fluid collections, celiac plexus block and neurolysis, while diagnostic EUS-like CE-EUS (20%) and real-time elastography (12%) were also conducted. Most complications occurred during the first 7 days after EUS-FNA/FNB or pseudocyst drainage. EUS and the additional tools have high technical success rates and low rates of complications. The EUS methods are safe, cost effective and indispensable for the diagnostic or therapeutic management in gastroenterological everyday practice.

## 1. Introduction

Endoscopic ultrasound (EUS) was introduced as a pure diagnostic imaging method in 1980 when an ultrasonography probe was attached to an endoscope. Since then, it has revolutionized the approach to gastrointestinal (GI) endoscopy in 21st century [1]. EUS gained wide acceptance not only as a diagnostic tool of primary or metastatic tumors [2,3,4,5], but also for GI cancer staging and as a minimally invasive therapeutic approach to mediastinal, intra-luminal and extra-luminal GI tract malignancies [1,6]. It allows a detailed analysis of the parietal wall, abdominal/mediastinal lymph nodes [7,8] and surrounding organs in otherwise difficult anatomic locations. Since its introduction, EUS has undergone substantial technological advances. A key advantage of this procedure is the ability to collect tissue from a lesion suspected of malignancy through the use of EUS-guided fine-needle aspiration (EUS-FNA).

The first EUS-FNA in the GI tract was reported in 1992 by Vilmann et al. [9]. The same group developed the first needle device for EUS-FNA, and all other commercially available needles have striking similarities with it [10]. In the same year, EUS-FNA of both upper and lower GI tract were reported [11,12]. The common sites for EUS-FNA are pancreas, bile duct, liver, lymph nodes at various mediastinal and abdominal sites, suspicious GI tract wall thickening or submucosal lesions, adrenal glands, perirectal lesions, retroperitoneal masses, posterior mediastinum, and central pulmonary masses [13].

EUS-FNA is generally a safe technique and complications are rare and usually self-limiting: perforation (0.2–0.8%), bleeding from the needle insertion site (0.13–0.69%), pancreatitis (0.44–0.92%) and infection, which accounts for 0.4–1.7% of cases [14]. Related morbidity and mortality rates are less than 1% (0.98% respectively 0.02%) [15]. For the purpose of reducing the complication rates, prior to performing EUS-FNA it is imperative to exclude pre-existing hematological conditions such as: coagulopathy (INR > 1.2), thrombocytopenia (platelets < 100,000) and chronic or recent intake of thienopyridines (e.g., clopidogrel) [16]. The risk factors for perforation are represented by inexperienced operators or the involvement of trainees, geriatric patients, esophageal cancer, previous difficult intubation, or cervical osteophytosis [14].

The aim of this study was to extensively assess in dynamic the EUS experience of our tertiary referral centers following several aspects within regular activity. We performed not only a population study and an overall assessment of the EUS GI procedures (with a particular focus on identifying the potential complications), but we also evaluated the performance of EUS-FNA versus EUS-FNB in diagnosing pancreatic neoplasms. Additionally, we investigated the contribution of associating contrast-enhanced ultrasound imaging (CE-EUS) with EUS-FNA/FNB for differentiating solid pancreatic lesions or cystic pancreatic lesions.

## 2. Materials and Methods

### 2.1. Study Design

This study is a retrospective analysis of a prospectively maintained database of patients who underwent EUS for the evaluation of benign and malignant diseases of the upper/middle/lower GI tract and of the organs in its proximity. Before each EUS procedure, medical data were recorded such as: patient demographics, referral details and indications, presumptive diagnosis, management plan, endosonographic features, technical success, presence of adverse events/complications. Patients from our institutional database who underwent CE-EUS and EUS-FNA/FNB for the evaluation of a pancreatic lesion were further retrieved. EUS-FNA/FNB data included details regarding needle type and size, lesion site and size, number of passes, cytological and histological diagnosis. Additionally, we assessed the benefit of CE-EUS in describing pseudocysts and the benign/premalignant/malignant character of pancreatic cystic lesions, as well as targeting the viable component of solid pancreatic masses (Figure 1).

### 2.2. Ethical Statements

The study protocol was approved by the institutional research ethics board and was conducted according to the principles of the Declaration of Helsinki. All patients gave their written informed consent before undergoing the EUS procedures.

### 2.3. Study Population

Patient inclusion criteria included the following: (1) age ≥18 years, (2) a GI lesion detected by at least 1 imaging modality, (3) the patient provided informed consent.

Patient exclusion criteria included the following: (1) endoscopy was impossible to perform, (2) failure of other organs, (3) hematologic instability (high risk of bleeding), (4) anesthesia allergy.

### 2.4. EUS Equipment and Techniques

EUS examinations were performed using linear echoendoscopes (GF-UCT 260, Olympus Medical Systems, Tokyo, Japan or EG 3870 UTK Pentax Medical Corporation, Tokyo, Japan) coupled with the corresponding US processor (Aloka Prosound Alpha-10, Aloka, Tokyo, Japan or Hitachi Preirus, Hitachi Medical Corporation, Tokyo, Japan). Commercially available 22 and 25-gauge FNA needles (EZ Shot 2 and EZ Shot 3 Plus Nitinol, Olympus, Tokyo, Japan; Expect, Boston Scientific, Natick, MA, USA; EchoTip Ultra, Cook Medical, Bloomington, IN, USA) or FNB needles (Acquire, Boston Scientific; EchoTip ProCore, Cook Medical) were used.

### 2.5. CE-EUS Procedure

CE-EUS was performed using the above-mentioned ultrasound systems with the patients under propofol sedation. Once the lesion was confirmed using the B-mode, the microvascularization was evaluated over 2 min after intravenous injection of the contrast agent (4.8 mL SonoVue, Bracco, Milan, Italy), followed by flushing with 10 mL saline solution. The solid pancreatic lesion enhancement and intensity were compared with the adjacent parenchyma and was classified as hypervascular, isovascular or hypovascular during both arterial and venous phase. The CE-EUS criteria used for labeling the pancreatic cysts were: cystic wall size, presence/absence and the size of mural nodules, presence/absence of calcifications, Wirsung duct enlargement, the presence/absence of contrast enhancement in the cystic wall and cystic septae. The set-up for exposing the cystic lesion was a low mechanical index and gain. The mechanical index was set up between 0.1 and 0.2, while gain was adjusted to lowest levels in order to avoid tissue signal. The viable tissue of solid formations was punctured post-contrast with the avoidance of necrosis areas. For cysts, the septae or wall nodules were punctured if displaying a contrast enhanced signal.

### 2.6. EUS-FNA/FNB Sampling Procedure

Procedures were carried out under deep propofol sedation. After sedating the patient, fundamental B mode EUS assessment was performed in order to identify the GI lesion. Doppler imaging was employed to detect the vessels interfering with the puncture line. After the puncture, the endosonographer verified that the tip of needle was properly positioned within the target lesion; then the stylet was completely removed and a 20 mL syringe was mounted on the handle of the needle. Once the needle was safely inserted into the lesion, the periphery of the tumor was targeted and the needle passes for EUS-FNA/FNB sampling were performed using the fanning technique with high negative pressure. The suction generated by the negative pressure eased the pass of tissue into the needle. Samples were dispossessed by reintroducing the stylet into the needle.

After the EUS-FNA sampling procedure, 2 needle passes were expelled onto glass slides for cytological examination, another 2 needle passes were expelled into formalin containers for cell block and 1 needle pass was used for DNA/RNA assessment. Rapid on-site evaluation (ROSE) was not performed.

With regard to EUS-FNB sampling procedure, 2 needle passes were used for cell block and 1 needle pass for DNA/RNA assessment. Macroscopic on-site evaluation (MOSE) was performed for the EUS-FNB samples which were confirmed as adequate by visually inspecting for the presence of whitish specimens in the expelled tissue. Needle passes were repeated until the endosonographer considered that an adequate sample had been obtained. If sufficient tissue had been acquired the procedure was completed within 2 needle passes. Additionally, when insufficient specimens had been acquired, up to 5 needle passes were completed.

### 2.7. Histologic Assessment

The EUS-FNA/FNB samples were subjected to cytologic/histologic analysis. Firstly, we assessed the adequacy of the samples for histological diagnosis. A sample was considered “inadequate” if the tissue obtained by puncture had an insufficient quantity. A sample was defined as malignant if it was either positive or suspect for malignancy. A negative or atypical sample was considered benign. The pathologists involved in this study have experience of over 15 years in performing cytological and histological EUS-FNA/FNB examinations.

### 2.8. Final Diagnosis

Final diagnosis was based on histopathology of surgical specimens or EUS-guided tissue acquisition. Among patients in whom a confirmed diagnosis could not be obtained through FNA/FNB sampling or whose lesion was not surgically resected, clinical follow-up and imaging studies (ultrasonography, EUS, computed tomography, magnetic resonance imaging) were conducted for at least 6 months after the endoscopic procedure. Malignancy was confirmed in cases where lesion progression and/or metastasis was observed on follow-up imaging, whereas benign disease was confirmed in cases with a stable lesion without imaging features of malignancy and without increasing size or metastasis during follow-up.

### 2.9. Study Outcomes

**Aim**: to extensively review in dynamic the EUS experience of our tertiary referral centers


**The primary endpoint of the study**


-to assess the complication rate of EUS-guided diagnostic and interventional procedures, including EUS-FNA/FNB


**The secondary endpoints to investigate:**
-the EUS-FNA/FNB sensitivity and specificity for the diagnosis of pancreatic lesions-the contribution of associating CE-EUS with EUS-FNB for differentiating solid pancreatic lesions without on-site cytopathology-the entanglement of CE-EUS in describing pseudocysts and the benign/premalignant/malignant character of pancreatic cystic lesions-the diagnostic sensitivity of CE-EUS guided FNA/FNB sampling and conventional EUS-guided FNA/FNB sampling of solid pancreatic masses


### 2.10. Statistical Analysis

Student *t*-test was used for the comparison of continuous variables as mean with standard deviation. Categoric variables were studied as percentage. *p* values of <0.05 were considered statistically significant. All authors had access to the study data and reviewed and approved the final manuscript.

## 3. Results

### 3.1. 20 Years of EUS Study Group

A total of 2935 patients undergoing EUS between January 2002 and December 2021 (20 years) were included. The mean age was 58, female to male ratio 1:2, with 65% of the patients residing in an urban area. The demographic characteristics of the study population were highly heterogeneous with regard to age, gender, family status, income, education level, and occupation. Most of the patients were referred from gastroenterologists (73%) and oncologists (15%) for EUS-FNA/FNB in order to establish the final diagnosis before initiating the oncological regimes and 22% of the patients were referred by surgeons, internists, pulmonologist, or radiologists. Procedures were carried out under deep propofol sedation. A total of 1883 diagnostic EUS and 1052 interventional EUS procedures consisting of 840 EUS-FNA and 212 EUS-FNB were performed.

Therapeutic procedures performed during our study included the subsequent interventions: endoscopic transmural drainage of pancreatic fluid collections, with placement of either plastic or metal stents in 40% of the cystic lesions, celiac plexus block and neurolysis in under 1% of all EUS cases (Table 1). Contrast enhanced-EUS (20%) and real time elastography (12%) were conducted in some of the cases.

According to the lesion localization, EUS procedures targeted the following locations: pancreatobiliary, esophageal and gastric/duodenal, mediastinum and lungs, liver, colorectal, and retroperitoneal (Figure 2). A description according to the location and indication of EUS procedures is shown in Table 2. The most common indications for EUS were choledocholithiasis and malignancy work-up.

Technical difficulties encountered were correlated to unpassable luminal strictures that prolonged the duration or terminated the procedure. Regarding the length of the EUS procedures, the shortest time was spent during exploratory EUS whereas the most time-consuming were the interventional procedures, especially when associated with CE-EUS and elastography evaluations (Table 3).

Most complications occurred during the first 7 days after EUS-FNA/FNB or pseudocyst drainage. A total of 50% of these patients recovered with conservatory therapy whereas 33% required surgical intervention (Table 4). A total of 20% of the patients who underwent EUS drainage of a pancreatic cyst received wide-spectrum antibiotics for 3 days, on average. Two deaths were registered during the first week after the EUS procedures. In both cases, the cardiovascular mortality was associated with an ASA (American Society of Anesthesiology) score of 3/4. The post-endoscopic procedure complications were further classified according to Clavien et al. [17] (Table 5). There was no relevant statistical correlation between the type of FNA/FNB needle and the complications.

### 3.2. EUS-FNA/FNB + CE-EUS of Solid Pancreatic Lesions Subgroup

264 patients who underwent EUS-FNA/FNB associated with CE-EUS for the evaluation of a solid pancreatic lesion were enrolled in the study. CE-EUS was firstly used in our institution in 2008, whereas FNB needles were used in more than 85% of cases after 2018.

The pancreatic masses where located as it follows: 2/3 at the head level and 1/3 at the level of the body and tail. The size of the pancreatic masses ranged from 10 to 78 mm, with an average diameter of 35 mm. For obtaining the core tissue, fanning technique was applied with a mean number of needle passes of 2 and a range from 2 to 5 passes. Final pathology revealed pancreatic ductal adenocarcinoma–PDAC (149) (Figure 3), mass-forming pancreatitis–MFP (57) (Figure 4), pancreatic neuroendocrine tumors–pNETs (23) (Figure 5 and Figure 6), undifferentiated carcinoma (17), mucinous carcinoma (5), and pancreatic metastasis (5).

In 48% of the cases, the pancreatic lesion was described as hypoenhancing, with a final diagnosis of malignancy. Regarding the enhancement patterns, hypovascularity in both arterial and venous phase was associated with PDAC, hypervascularity or isovascularity in both phases were associated with either MFP or NETs, whereas the carcinomas were hypervascular (Table 6). A heterogeneous appearance with non-enhancing areas was noted in a small percentage of the hypoenhancing lesions and it might suggest necrosis. The overall diagnostic accuracy was 91%.

### 3.3. EUS-FNA/FNB + CE-EUS of Pancreatic Cystic Lesions Subgroup

In all, 152 patients (age 65 ± 10 years, female = 61, male = 85) with cystic lesions of the pancreas underwent CE-EUS assessment. The visualization of the cystic wall and inner parts was possible in all cases. A total of 72% of the patients presented lesions with specific characteristics for pancreatic pseudocysts (no visible wall, septae or nodule vascularization) and were followed up only for the purpose of the study. None of these cases displayed a malignant progression during the 12 months of follow up, a slight increase in the cyst was observed in three patients (under 10 mm) and no further action was required. A total of, 28% of the patients were categorized as suspected cystic lesions (serous cystadenomas, intraductal papillary mucinous neoplasm, mucinous cystic neoplasm, etc.) and further underwent EUS-FNA/FNB procedure for the puncture of cystic wall/septae/nodule. In 61 cases a metal or plastic stent was placed into the pseudocyst for drainage.

The CE-EUS criteria used for labeling the cystic pancreatic tumors were cystic wall size > 3 mm, Wirsung duct enlargement, the presence of contrast enhancement in the cystic wall (Figure 7), septae (Figure 8), the presence and the size of mural nodules (Figure 9).

A majority of the malignant cysts were diagnosed by EUS-FNA/FNB, as the biopsy was performed under constant visualization and engaged towards the hyperenhancing components of the cysts. CE-EUS was significantly more accurate than the standard EUS in diagnosing malignant cysts (88% and 59%, respectively, *p* < 0.05). Furthermore, CE-EUS allowed the differentiation of mural nodules from mucous clots with a sensitivity of 100% and 84% specificity.

### 3.4. EUS-FNA versus EUS-FNB of Solid Pancreatic Lesions Subgroup

In all, 1345 patients were referred to our institution for the investigation of a solid pancreatic lesion. The patients who underwent EUS-FNA/FNB for the assessment of a solid pancreatic lesion were further retrieved from the total number of 1052 cases of EUS-FNA and EUS-FNB. Hence, 665 patients were counted in the FNA group and 212 patients in the FNB group. There was no statistically significant difference between the groups in terms of age (median age 61 years old), sex (male to female ratio 2:1), lesion size (34.6 mm for the FNA group and 37.4 mm for the FNB group), or puncture method (fanning technique). The median procedure time was longer in the EUS-FNA group than in the FNB group, 18 min and 13 min, respectively. In the FNA group the lesions where equally located on the head and body of the pancreas, whereas in the FNB group the pancreatic mass location was 2/3 head, 1/3 body and tail. An increasing preference for the use of biopsy needles was noted year after year, with EUS-FNB reaching 85% of the cases in 2021. The patients from the FNB group obtained a pathological diagnosis within two needle passes, while the FNA group required an average of three needle passes (with a range of one to five needle passes). For the EUS-FNA/FNB, five types of 22/25 G puncture device were used over the years, but two main needles were in routine use: Acquire 22 G and EZ shot2 22 G (Table 7).

In both groups, pancreatic ductal adenocarcinoma was the most common disease, followed by mass-forming pancreatitis and pancreatic neuroendocrine tumors; whereas undifferentiated carcinoma, mucinous carcinoma and pancreatic metastasis were diagnosed to a lesser extent. The parameters of diagnostic performance of EUS-FNA versus EUS-FNB are described in Table 8.

## 4. Discussion

In this study, we examined the patient characteristics and diagnostic abilities of EUS-FNA, EUS-FNB and CE-EUS. With a diagnostic accuracy ranging between 77% and 95% for sampling pancreatic masses, EUS-FNA is currently the standard of care [3]. EUS-FNA yields a particularly high sensitivity (85–89%) and specificity (96–99%) for differentiation between a benign or malignant lesion [18]. Regarding the differential diagnosis of pancreatic neoplasms, various factors influence the diagnostic accuracy, among which the most significant are the size and location of the mass, concomitant chronic pancreatitis, and the presence of peritumoral desmoplastic stromal reaction [19,20,21]. Furthermore, taking into consideration strict cytological criteria, EUS-FNA sensitivity reached values as low as 77%, even in experienced hands, as a consequence of inadequate samples, the presence of fibrosis or extensive necrosis reaction [22,23]. Therefore, rapid on-site evaluation (ROSE) by a cytopathologist has been proposed to improve EUS-FNA diagnostic accuracy by confirming the presence or absence of tissue samples during examination [24,25,26]. Initially described by Hikichi et al. [27], ROSE allows the assessment of sample adequacy/cellularity and, consequently, increases the overall diagnostic accuracy rate and reduces the number of needle passes. Although the real impact of ROSE regarding these aspects is not well established, its introduction in clinical practice is desirable. Unfortunately, ROSE is not widely available and currently cannot be introduced at numerous facilities because the presence of a cytopathologist in the endoscopy room is limited for some practical reasons: the long inspection time, local shortage of human resources, or high costs [26,28]. A similar scenario is the reason for ROSE not being available in our institution, hence a low sensitivity of EUS-FNA for pancreatic tumors was noted in our study (75.9%).

Several methodological developments have been proposed for improving the diagnostic yield of EUS-FNA, such as changing the needle tip, using larger-gauge needle, increasing the number of passes within the lesion, and adjusting the tissue sampling procedure (stylet slow-pull versus standard suction, dry versus wet technique, fanning method versus only moving “to and fro’’) [5,29,30]. On the other hand, since EUS-FNA is an operator-dependent technique that has a long learning curve, it is generally recognized that the diagnostic accuracy increases with operator experience [31,32].

Although EUS-FNA is usually adequate for the final diagnosis of pancreatic ductal adenocarcinoma, it does not contain the stroma, a preserved architecture being essential for a positive diagnosis of other pancreatic solid tumors and benign conditions. The shortcomings of cytological samples obtained through EUS-FNA are represented bythe following aspects: not allowing immunohistochemistry, phenotype analysis, or gene mutation identification [33]. Endoscopic ultrasound tissue acquisition, in the form of fine needle biopsy (EUS-FNB) was designed to overcome these limitations in order to provide a proper quantitative core tissue. This allows the assessment of preserved histologic architecture, further immunohistochemical staining, and gene profiling, which are essential factors for accurate risk stratification and personalized oncological management. With EUS-FNB, adequate tissue sampling can be expected with a small number of punctures and without ROSE [34]. The first original FNB needle (QuickCore^®^ Biopsy Needle; Cook Medical) was introduced in the 2000s and it was a Tru-Cut needle. Currently, several core biopsy needles are available on the market and can be counted as non-cutting or cutting type, including side-type or end-type. In our study, the Acquire FNB needle was used with success in 92% of the punctured cases. Total procedural time was reduced by performing FNB with MOSE. The decreased number of passes required by EUS-FNB versus EUS-FNA diminished the anesthesia time and costs as well. EUS-FNB was employed as a salvage technique subsequent to unsuccessful FNA.

EUS-FNA/FNB is complemented by contrast-enhanced endoscopic ultrasonography (CE-EUS) in depicting malignant lesions by identifying the target for the puncture, allowing the evaluation of vascularity and clearly showing the outline of the lesion [35]. The clinical implications of CE-EUS reach a wide range of pathologies: differential diagnosis of focal pancreatic masses, the evaluation of cystic lesions (specifically allowing the distinction of mucous plugs over small neoplastic nodules), the assessment of acute or chronic pancreatitis and its complications, lymph-node classification, evaluation of gallbladder or intraductal biliary masses, subepithelial lesions, or description of pancreatic filling defects [36]. Similar to previous research, in our study, CE-EUS provided supplementary information on the vasculature of lesions and the relationship with the vessels surrounding the lesion. Also, it helped with the identification of the EUS-FNA/FNB target area among distinct pathological regions.

As a general overview, the population demographic data of our study display the heterogeneity of different gastrointestinal lesions, widely ranging from benign to malignant. This highlights the importance of EUS in suspected gastrointestinal pathologies. A majority of the EUS was carried out for pancreatobiliary indications (66%) followed by esophageal and gastric/duodenal (13%) indications, similar to the study of Shil et al. [37]. The pancreatobiliary lesions were common in the study of Chong et al. [38] while the esophageal indication was commonest in the study of Kalade et al. [39]. Just as in the Australian study [39], the most common referrals for EUS came from gastroenterologists followed by oncologists, surgeons, internists, pulmonologists, and radiologists. In our experience, a limited number of complications were encountered. A total of 50% of the complications were graded as III-b according to Clavien-Dindo classification. Two deaths were registered due to myocardial infarction in the case of two patients with a pre-endoscopy critical status. In the literature, complication rates of EUS-FNA are less than 0.1–1% [40]. Owing to the rather small representation of complications in the dataset, regression analysis did not fit our study, although it would have been helpful to better understand which are the risk factors of morbidity and mortality.

This study has several limitations. Firstly, it is a retrospective dual-center report. Therefore, future studies should inspect the characteristics and diagnostic abilities of EUS-FNA/FNB in a multi-center setting. Secondly, the number of EUS-FNB cases was less than the number of EUS-FNA procedures. Hence, upcoming research with substantial case numbers would be beneficial.

## 5. Conclusions

This is the first report of a dual-center EUS experience over the course of the past 20 years. EUS and the additional tools have high technical success rates and low rates of complications. The EUS methods are safe, cost effective and indispensable for the diagnostic or therapeutic management of digestive diseases. The introduction of EUS-FNB with a new generation of needle showed a similar high safety profile to EUS-FNA; however, EUS-FNB requires a lower number of needle passes for the acquisition of a proper tissue sample. Considering that both EUS-FNA and EUS-FNB ensured a comparable diagnostic yield, without rapid on-site evaluation, the preference of one technique over the other should be centered on local availability and expertise. CE-EUS is a valuable method to distinguish malignant cysts from pancreatic pseudocysts. CE-EUS allows detailed visualization of the dynamic enhancement patterns, hence it helps to identify the target of EUS-FNB among different pathological areas of the lesions. CE-FNB-EUS can be used for the differential diagnosis and adequate sampling of solid pancreatic lesions without on-site cytopathology. However, it is still not superior to pathological diagnosis and further studies are required in order to establish more specific parameters, the proper associations with other imagistic techniques and, eventually, with non-invasive procedures.

## Figures and Tables

**Figure 1 diagnostics-12-01641-f001:**
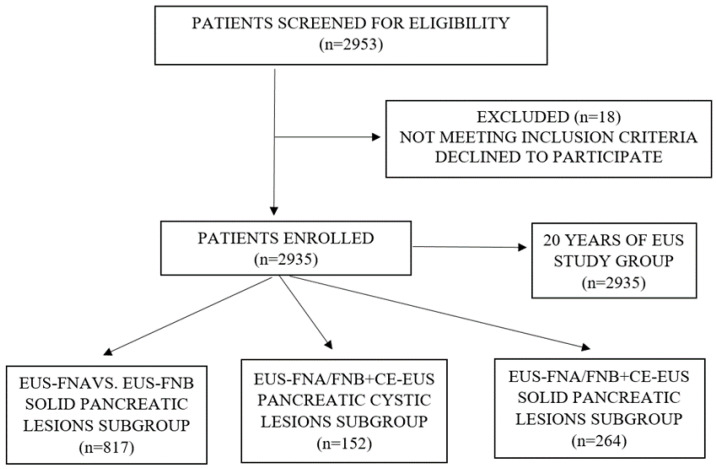
Study design.

**Figure 2 diagnostics-12-01641-f002:**
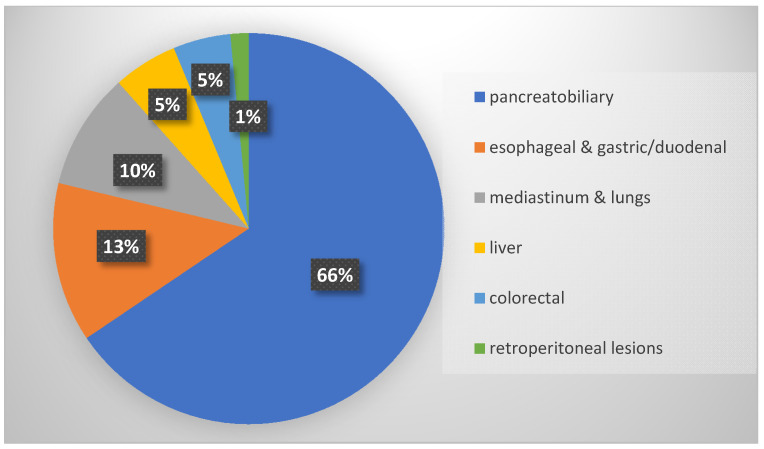
EUS assessment (%) with regard to lesion location.

**Figure 3 diagnostics-12-01641-f003:**
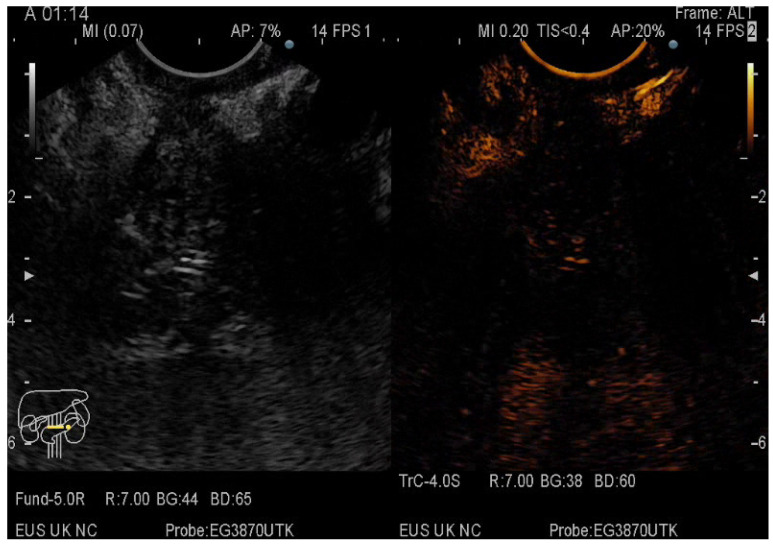
CE-EUS image of a PDAC showing a hypoenhancing solid mass in both arterial and venous phase.

**Figure 4 diagnostics-12-01641-f004:**
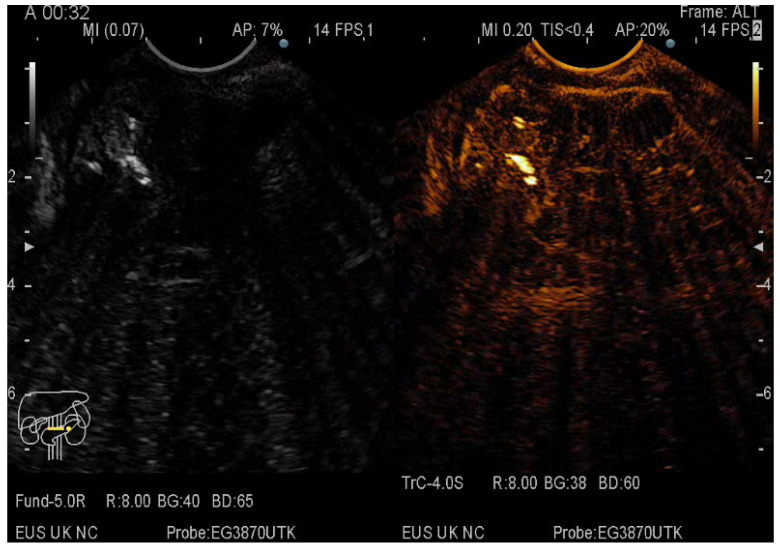
CE-EUS image of a MFP revealing a solid mass with hyperenhancement in the arterial phase and no wash-out in the venous phase.

**Figure 5 diagnostics-12-01641-f005:**
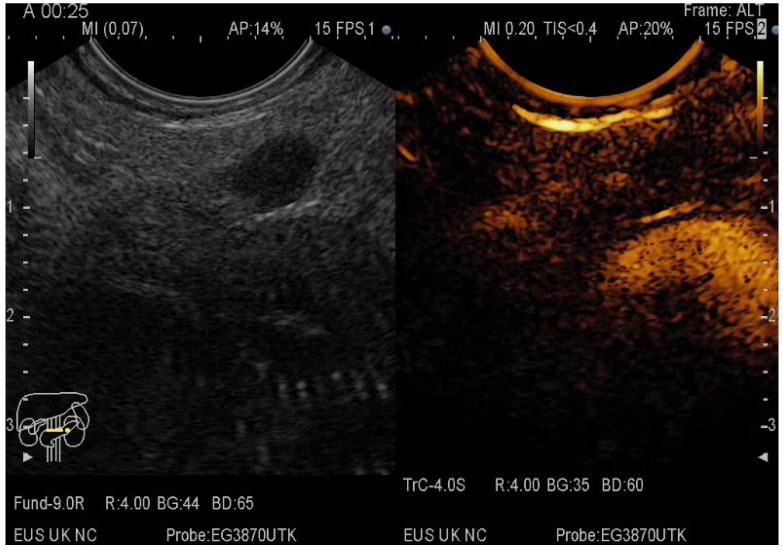
CE-EUS image of a pNET revealing an isoenhancing solid mass in the arterial phase and a discrete wash-out in the venous phase.

**Figure 6 diagnostics-12-01641-f006:**
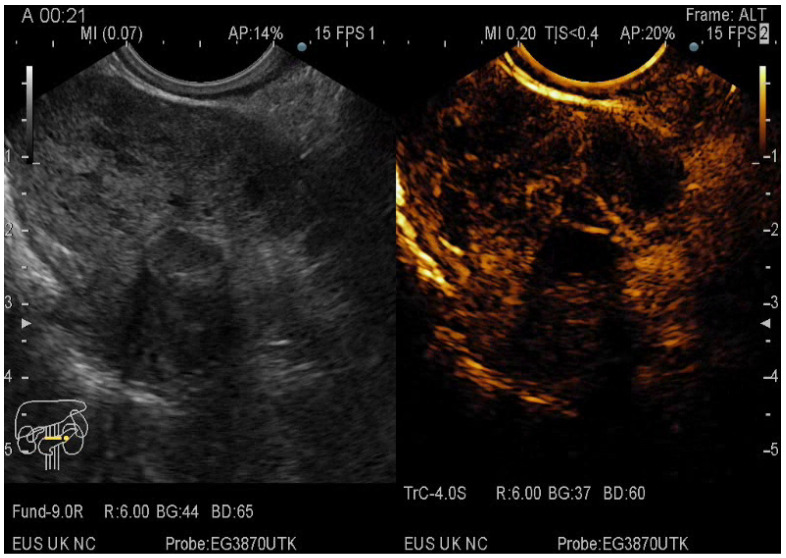
CE-EUS image of a pNEC (pancreatic neuroendocrine carcinoma) with aspect in the arterial phase and wash-out in the venous phase.

**Figure 7 diagnostics-12-01641-f007:**
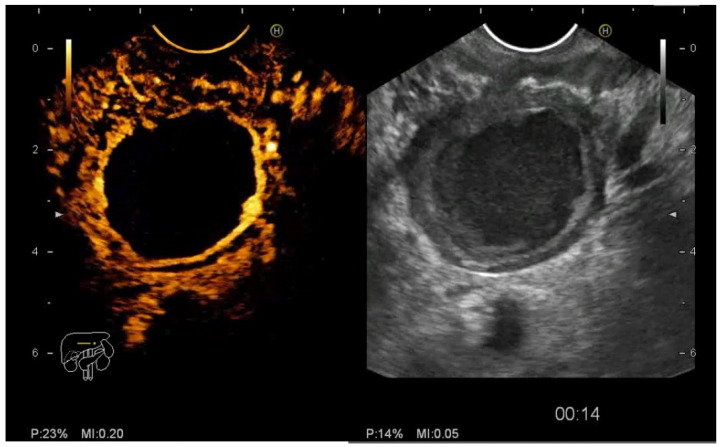
Contrast enhancement of the cystic wall.

**Figure 8 diagnostics-12-01641-f008:**
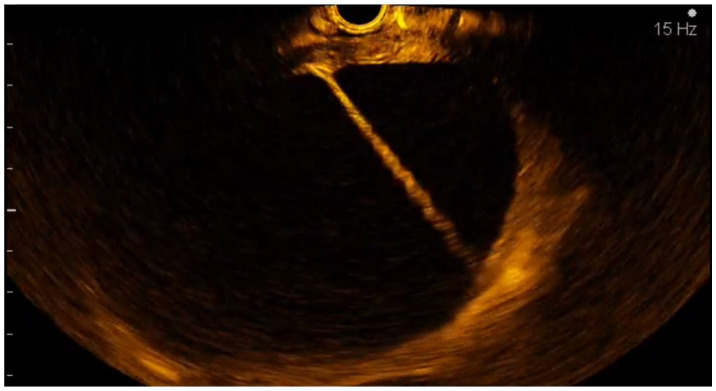
Contrast enhancement of a cystic septae.

**Figure 9 diagnostics-12-01641-f009:**
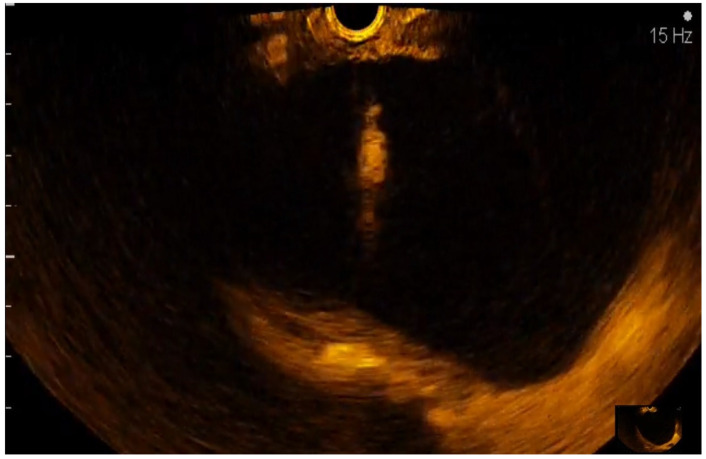
Contrast enhancement of a cystic mural nodule.

**Table 1 diagnostics-12-01641-t001:** Diagnostic and interventional EUS procedures performed during the two decades.

	EUS-Diagnostic	EUS-FNA/FNB Overall	Pancreatic EUS-FNA/FNB	Mediastinal EUS-FNA/FNB	Drainage of Pancreatic Fluid Collections	Celiac Plexus Block and Neurolysis
Percentage of all EUS cases	64%	36%	30%	5%	4%	<1%

**Table 2 diagnostics-12-01641-t002:** Location and indication of EUS procedures.

	Lesion Localization	Indication	No. of Patients
EUS procedures	pancreatobiliary	Assessment of the pancreas and ampullary lesions; pancreatic fluid collections drainage; biliary drainage; FNA/FNB of pancreatic cystic and solid lesions; cancer pain relief (celiac plexus neurolysis)	1937
	esophageal and gastric/duodenal	Assessment and FNA/FNB of esophageal subepithelial lesions, paraesophageal lymph nodes, gastric subepithelial lesions, intra-abdominal lymphadenopathy, duodenal subepithelial lesions, metastatic lesions; Staging of esophageal, gastric/duodenal malignancy; Assessment of esophageal and gastroesophageal varices	381
	mediastinum and lungs	Assessment and FNA/FNB of paraesophageal and mediastinal masses	293
	liver	Assessment of the left lobe of the liver	147
	colorectal	Staging of colorectal malignancy; Assessment and FNA/FNB of rectal subepithelial lesions, pelvic lesions and pelvic lymphadenopathy, Assessment of anal sphincter, Crohn disease fistulae	147
	retroperitoneal	Assessment of retroperitoneal lymph nodes and masses.	30

**Table 3 diagnostics-12-01641-t003:** EUS procedural time.

EUS Procedure	Exploratory	Interventional	Interventional + CE-EUS	Interventional + CE-EUS + Elastography
Procedure time (min) median, range	10.5 (5–15)	15.8 (10–45)	18.8 (13–50)	22.3 (16–55)

**Table 4 diagnostics-12-01641-t004:** Complications associated with the EUS procedures.

Complications	No.	FNA	FNB	Pseudocyst Drainage	WOPN	Conservative Treatment	Surgery	Death
Mild acute pancreatitis	2	Small solid tumors	-	-	-	x	-	-
Retroperitoneal bleeding	1	Neuroendocrine tumor	-	-	-	x	-	-
Subcapsular hematoma	1	Spleen	-	-	-	x	-	-
Peritonitis consequent to abscess	1	Malignant celiac trunk ganglia	-	-	-	-	X	-
Abscess	1	Cystic tumor, tail of the pancreas (IPMN)	-	-	-	-	X	-
Myocardial infarction	2	-	Advanced pancreatic tumor with peritoneal carcinomatosis	-	-	-	-	x
		-	-	-	HotAxios stent, severe acute biliary pancreatitis	-	-	x
Biliary peritonitis	1	-	-	x	-	-	X	-
Significant bleeding	1	-	-	x	-	-	X	-
Superinfected pseudocysts	2	-	-	x	-	EUS reintervention	-	-

**Table 5 diagnostics-12-01641-t005:** Complications grading according to Clavien-Dindo classification [17].

Grade	Grade Description	Complications	No. of Patients
I	Any variation of the patients post-endoscopic status without the need of pharmacological, surgical, endoscopic and radiological interventions. Acceptable pharmacological treatment: antiemetics, antipyretics, analgetics, diuretics and electrolytes.	Mild acute pancreatitis	2
II	Blood transfusions, therapeutic regimens other than those for grade I complications were required.	Retroperitoneal bleeding, subcapsular hematoma	2
III	Surgical interventions or endoscopic re-interventions were required		
III-a	Intervention not requiring general anesthesia	-	0
III-b	Intervention requiring general anesthesia	Peritonitis, abscess, significant bleeding	6
IV	Life-threatening complications		
IV-a	Single organ dysfunction (including dialysis)	-	0
IV-b	Multi organ dysfunction	-	0
V	Death of a patient	Myocardial infarction	2

**Table 6 diagnostics-12-01641-t006:** Contrast enhancement patterns of the main pancreatic lesions. Values are number of patients (%).

	PDAC	MFP	NETs	Carcinomas
Hyperenhancement	21 (14%)	45 (79%)	17 (74%)	22 (100%)
Hypoenhancement	127 (86%)	0	0	0
Isoenhacement	0	12 (21%)	6 (26%)	0

**Table 7 diagnostics-12-01641-t007:** Percentage of EUS FNA/FNB needles used.

	Acquire 22 G	EZ shot2 22 G
EUS-FNA	N/A	85.7%
EUS-FNB	92.6%	N/A

**Table 8 diagnostics-12-01641-t008:** Sensitivity, specificity, accuracy of EUS-FNA vs. EUS-FNB.

Pancreatic Tumors	EUS-FNA	EUS-FNB
Sensitivity (%)	75.9	86.8
Specificity (%)	100	100
Diagnostic accuracy (%)	84.6	90.5

## Data Availability

All data supporting the study can be located in the archive of: Research Center of Gastroenterology and Hepatology Craiova, Romania and Ponderas Hospital, Bucharest, Romania.

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
