# Peer review of "Technical Performance, Overall Accuracy and Complications of EUS-Guided Interventional Procedures: A Dynamic Landscape"

_diagnostics, 2022, doi:10.3390/diagnostics12071641_

Round 1

Reviewer 1 Report

In this manuscript, the authors reported the performance of EUS-guided FNA and FNB in diagnosing neoplasms. With the comparison of CE-EUS and EUS in differentiating solid and cystic pancreatic lesions, the authors concluded that EUS leads to high success rates and low complication rates. Overall the study is well-designed with results directly supporting the conclusions. The study is important to improve our current understanding and usage of EUS methods. Thus, the manuscript is suggested to be accepted in the journal. 

Author Response

Dear reviewer,

Thank you for your time and for your positive assessment of our paper!

Best of wishes!

Reviewer 2 Report

Dear Author,
I had the pleasure of rewiving your retrospective descriptive bi-centric study for the evaluation of the role of EUS and contrast-enhanced ultrasound imaging (CE-EUS) respectively in the management of lesions located in the gastrointestinal tract and adjacent organs and in differential diagnosis, over the last 20 years.
A total of 2935 patients a, including 1880 diagnostic EUS and 1052 EUS-FNA / FNB (80% FNA and 20% FNB).
The primary endpoint of the study is to evaluate the complication rate of EUS-guided diagnostic and interventional procedures, including EUS-FNA / FNB .
The secondary endpoints are the sensitivity and specificity of EUS-FNA / FNB for the diagnosis of pancreatic lesions and the contribution of EUS-FNA to EUS-FNB for differentiating solid pancreatic lesions from cystic lesions.
A matching of samples on histological diagnosis was performed.
The final diagnosis was based on post surgery histopathology or biopsy. In the remaining patients a clinical follow-up and imaging studies at least 6 months after the endoscopic procedure.

A total of 2935 patients who underwent EUS between January 2002 and December 2021 (20,188 years) were included. 1880 diagnostic EUS were performed and a total of 1052 EUS punctures consisting of 840 EUS-FNA and 212 EUS-FNB.

DESCRIPTIVE ANALYSIS
Major remarks
Concerning the study population:

Is there a difference in demographic characteristics between groups and within groups?
In order to better understand the potential heterogeneity of this large population, I would like to suggest the creation of a descriptive table according to the location and indication of the EUS

Concerning the main objective:

In order to make the reading of this retrospective descriptive study more fluid and to highlight your results, I would like to suggest that you create a table, taking into consideration the classification of morbidity and mortality, possibly at 30 days, of Clavien Dindo (Ann Surg 2009), or in major complications (>III for C/D or endoscopic or surgical procedure) and minor complications.

STATISTICAL ANALYSIS
Major remarks

In order to bring additional weight to the discrete results, and not to carry out exclusively a descriptive analysis, would it not be, without speaking of causality, to carry out a regression with the aim of better understanding which are the risk factors of morbidity and mortality and the diagnostic "failures". I would tend to force in your empty M0 models the location and the indication of the procedure.

Kind regards

Author Response

Dear reviewer, thank you very much for your valuable comments and recommendations! We managed to integrate most of it into the revised version of our manuscript.

Q1: Is there a difference in demographic characteristics between groups and within groups?
In order to better understand the potential heterogeneity of this large population, I would like to suggest the creation of a descriptive table according to the location and indication of the EUS

A1: Indeed, the demographic characteristics of the study population were highly heterogeneous. Following your suggestion, we added a descriptive table according to the location and indication of EUS.

A description according to the location and indication of EUS procedures is shown in Table 2. The most common indications for EUS were choledocholithiasis and malignancy work-up.

Lesion Localization

Indication

No. of patients

EUS procedures

pancreatobiliary

Assessment of the pancreas and ampullary lesions; pancreatic fluid collections drainage; biliary drainage; FNA/FNB of pancreatic cystic and solid lesions; cancer pain relief (celiac plexus neurolysis)

1937

esophageal & gastric/duodenal

Assessment and FNA/FNB of esophageal subepithelial lesions, paraesophageal lymph nodes, gastric subepithelial lesions, intra-abdominal lymphadenopathy, duodenal subepithelial lesions, metastatic lesions; Staging of esophageal, gastric/duodenal malignancy; Assessment of esophageal and gastroesophageal varices

381

mediastinum & lungs

Assessment and FNA/FNB of paraesophageal and mediastinal masses

293

liver

Assessment of the left lobe of the liver

147

colorectal

Staging of colorectal malignancy; Assessment and FNA/FNB of rectal subepithelial lesions, pelvic lesions and pelvic lymphadenopathy, Assessment of anal sphincter, Crohn disease fistulae

147

retroperitoneal

Assessment of retroperitoneal lymph-nodes and masses.

30

Q2: In order to make the reading of this retrospective descriptive study more fluid and to highlight your results, I would like to suggest that you create a table, taking into consideration the classification of morbidity and mortality, possibly at 30 days, of Clavien Dindo (Ann Surg 2009), or in major complications (>III for C/D or endoscopic or surgical procedure) and minor complications.

A2: According to your suggestion, we created a table with the classification of the complications encountered after the endoscopic procedures, taking into consideration the Clavien Dindo (Ann Surg 2009) classification.

Grade

Grade definition

Complications

No. of patients

I

Any deviation from the normal postoperative course without the need of pharmacological, surgical, endoscopic and radiological interventions. Acceptable pharmacological treatment: antiemetics, antipyretics, analgetics, diuretics and electrolytes.

Mild acute pancreatitis

2

II

Blood transfusions, therapeutic regimens other than those for grade I complications were required.

Retroperitoneal bleeding, subcapsular hematoma

2

III

Surgical interventions or endoscopic re-interventions were required

III-a

intervention not under general anesthesia

-

0

III-b

intervention under general anesthesia

Peritonitis, abscess, significant bleeding

6

IV

Life-threatening complications

IV-a

single organ dysfunction (including dialysis)

-

0

IV-b

multi organ dysfunction

-

0

V

Death of a patient

Myocardial infarction

2

Q3: In order to bring additional weight to the discrete results, and not to carry out exclusively a descriptive analysis, would it not be, without speaking of causality, to carry out a regression with the aim of better understanding which are the risk factors of morbidity and mortality and the diagnostic "failures". I would tend to force in your empty M0 models the location and the indication of the procedure.

A3: We have contacted our local statistics department and after parsing the database they came with the idea that regression analysis would not be fitted for our data. As regression aims to find the correlations between a predicted variable and one or more predictor variables, and in some way it is a generalizing model. Due to the rather small representation of complications in the dataset (e.g. only 2 post-procedural deaths were encountered), it is more likely that important information will be lost in the generalization. Additionally, the data variability within the group is quite large.

Reviewer 3 Report

Dear Editor,

Thank you for inviting me.

We suggest a linguistic revision and to expand the Discussion section.

The paper is very interesting and deserves publication.

Author Response

Dear reviewer,

Thank you for your time and for your valuable feedback!

We did follow your recommendations, hence a linguistic revision was performed and the Discussion section was expanded.

Best wishes!

Round 2

Reviewer 2 Report

Dear author, thanks for making the suggested changes. Your results are highlighted and you provide a description of the clinical practices carried out on a daily basis.

Best regards